# CaTS: Calibrated Test-Time Scaling for Efficient LLM Reasoning

**Chengsong Huang**[1]**, Langlin Huang**[1] **Jixuan Leng**[2]
**Jiacheng Liu**[3], **Jiaxin Huang**[1]
[1]Washington Univeristy in St. Louis
[2]Carnegie Mellon University [3]University of Washington
{chengsong,h.langlin,jiaxinh}@wustl.edu
jixuanl@cs.cmu.edu, liujc@cs.washington.edu

## Abstract

Increasing test-time computation is a straightforward approach to enhancing the quality of responses in Large Language Models (LLMs). While Best-of-N sampling and Self-Consistency with majority voting are simple and effective, they require a fixed number of sampling responses for each query, regardless of its complexity. This could result in wasted computation for simpler questions and insufficient exploration for more challenging ones. In this work, we argue that model confidence of responses can be used for improving the efficiency of test-time scaling. Unfortunately, LLMs are known to be overconfident and provide unreliable confidence estimation. To address this limitation, we introduce **Self-Calibration** by distilling Self-Consistency-derived confidence into the model itself. This enables reliable confidence estimation at test time with one forward pass. We then design **Calibrated Test-Time Scaling** (CaTS), adapting common repeated sampling methods, such as self-consistency and Best-of-N to handle queries of various difficulty. We also show that CaTS-SC is provably better than vanilla self-consistency. Experiments on three LLMs across nine datasets demonstrate the effectiveness of our approach. Specifically, applying confidence-based Early Stopping (CaTS-ES) to Best-of-N improves MathQA accuracy from 73.7 to 83.6 with a sample budget of 16 responses, demonstrating the effectiveness of the confidence-based sampling strategy at inference time. Our codes are available at https://github.com/Chengsong-Huang/Self-Calibration.

## 1 Introduction

Leveraging additional computation during inference can enhance the quality of responses generated by large language models (LLMs) (Snell et al., 2024a; Yao et al., 2023; Wu et al., 2024; Chen et al., 2025a). Among these methods, repeated sampling (Brown et al., 2024) such as Best-of-N (Cobbe et al., 2021) and Self-Consistency (Wang et al., 2022b) generate multiple candidate responses and select the final answer by a scoring model or a majority voting rule. While these methods have proven effective, they require a fixed amount of sampled responses for each query regardless of its difficulty and complexity. Although increasing the sample size generally improves performance, it also increases computational costs and inference time (Amini et al., 2024). This is

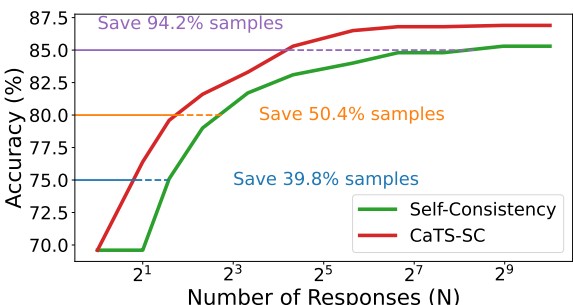

Figure 1: Accuracy over response numbers of vanilla Self-Consistency (SC) vs. CaTS-SC on MathQA using our trained Llama-3.1-8B-Instruct model. The horizontal lines mark the response usage difference required for CaTS-SC to reach the same accuracy with vanilla SC.

particularly inefficient for simple questions like "2 + 3 = ?", where a few samples are sufficient to find the correct solution (Chen et al., 2024), and extensive sampling is unnecessary.

Previous adaptive sampling strategies (Aggarwal et al., 2023; Li et al., 2024; Wan et al., 2024) typically design lightweight stopping criteria to determine whether additional responses should be sampled. However, they often incorporate manually designed features or heuristic rules, such as early stopping when the model generates the same response three times consecutively, which can limit their generalizability across different tasks and models. Therefore, it is critical to design a task-independent, model-agnostic approach without heavy reliance on human-designed heuristics.

We propose an efficient test-time sampling method by using model confidence for dynamic sampling adjustment, since confidence can be seen as an intrinsic measure that directly reflects model uncertainty on different tasks. However, extracting accurate confidence can be challenging since LLMs are known to be overconfident in their own responses (Lin et al., 2022; Xiong et al., 2023; Leng et al., 2024), and their confidences often exceed the actual accuracy, especially in small models. Self-Consistency (Wang et al., 2024a) can provide a relatively accurate confidence estimation by aggregating answer counts from multiple sampled solutions (Tian et al., 2023a), but it again requires sampling a large number of responses for each query beforehand.

To address this, we introduce **Self-Calibration** to train LLMs for accurate confidence estimation in only one forward pass, without requiring any human-labeled data. Specifically, we improve model calibration by distilling Self-Consistency-derived confidence into the model itself. This is done by constructing pseudo training tuples of query, answer, and confidence on a diverse training set. At test time, we propose **Calibrated Test-Time Scaling** (CaTS), including early stopping for Best-of-N when sampled responses reach a target confidence (CaTS-ES), and Self-Consistency weighted by confidence (CaTS-SC). We also provide a systematic proof of the conditions under which this confidence-weighted approach is theoretically guaranteed to outperform vanilla Self-Consistency.

Empirical experiments on three LLM architectures across nine datasets demonstrate that our confidence-based test-time scaling approaches consistently outperform their baseline counterparts under the same sampling budget. Specifically, both CaTS-ES for Best-of-N and CaTS-SC improve MathQA accuracy over their baselines from 73.7 to 83.6 with an average sampling budget of 16 responses. More importantly, our approaches can achieve comparable performance with substantially fewer computational resources. As shown in Fig. 1, CaTS-SC can save 94.2% samples to achieve an accuracy of 85.0, compared to standard Self-Consistency, demonstrating that reliable confidence estimation can significantly enhance the computational efficiency of test-time scaling.

## 2 REPEATED SAMPLING

Repeated sampling (Brown et al., 2024) is a framework that generates multiple responses with Chain-of-Thought prompting (Wei et al., 2022), then uses a verifier to get the final results. We will introduce three fundamental repeated sampling strategies, which aim to enhance response quality.

### 2.1 BEST-OF-N

For each input query $x$, multiple candidate responses $\{y_i\}$ are sampled, where $1 \le i \le N$. A scoring function—such as an additional reward model or a confidence generator—assigns each response a score $c_i = \text{Score}(y_i)$. The simplest selection strategy, known as Best-of-N (Cobbe et al., 2021), chooses the response with the highest score as the final answer.

### 2.2 SELF-CONSISTENCY

Self-Consistency (Wang et al., 2022b) (SC) selects the most frequent response among different candidates. Given candidate responses $\{y_1, y_2, \ldots, y_N\}$, the final answer is determined by majority voting. This approach enhances robustness by aggregating diverse model outputs rather than relying on a single highest-scoring response.

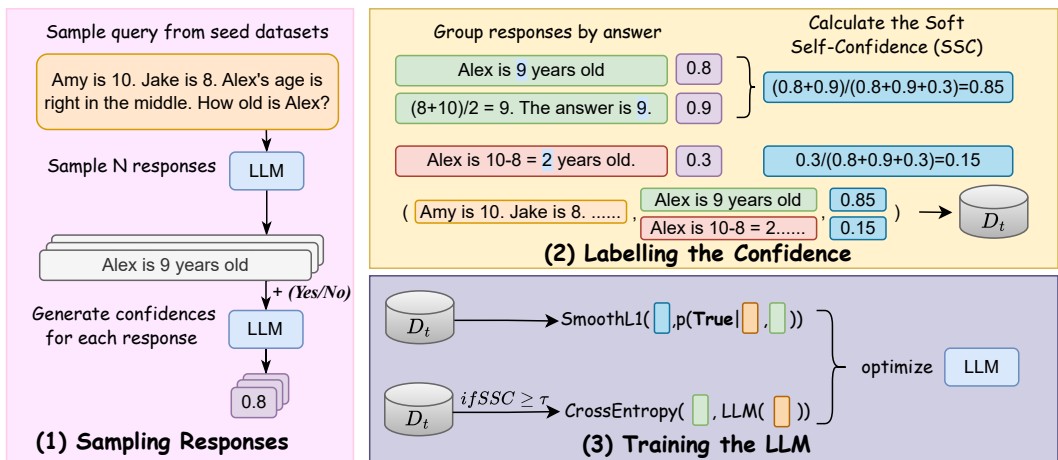

Figure 2: Illustration of the Self-Calibration framework. Given a query from the seed dataset, we sample $N$ responses from the LLM. We use a confidence querying prompt to let LLM assign a confidence score to each response. Responses are then grouped by their answers, and the Soft Self-Consistency (SSC) score is computed for each group. During training, all data tuples contribute to improving the model's calibration, while higher-confidence data is used to enhance the LLM's generation ability.

## 2.3 ADAPTIVE SELF-CONSISTENCY

Adaptive Self-Consistency (ASC) (Aggarwal et al., 2023) enhances the standard Self-Consistency approach by dynamically adjusting the number of samples based on agreement among generated responses. This method iteratively samples responses and calculates the cumulative frequency $v_k(z)$ and relative frequency $\hat{r}_k(z)$ of each unique answer $z$ after $k$ samples:

$$v_k(z) = \sum_{i=1}^{k} \mathbb{1}(y_i = z), \quad \hat{r}_k(z) = \frac{v_k(z)}{k}.$$

The sampling process continues until the maximum relative frequency $\hat{r}_k(z)$ exceeds a predefined threshold $\tau$. Formally:

$$\begin{cases} k \leftarrow k + 1, & \text{if } \max_z \hat{r}_k(z) < \tau, \\ y = \arg\max_z \hat{r}_k(z), & \text{otherwise.} \end{cases}$$

This adaptive strategy reduces computational costs by limiting the number of required samples while maintaining high accuracy in the final answer selection.

## 3 SELF-CALIBRATION

In this section, we provide an overview of our proposed Self-Calibration framework **without requiring any ground-truth answer**, illustrated in Fig. 2. First, we synthesize a set of input-output-confidence tuples $(x_i, y_i, c_i)$ from a seed dataset for training (Sec. 3.2). Using this synthetic dataset, we can train a language model with a combined loss to output calibrated confidence scores (Sec. 3.3).

## 3.1 CONFIDENCE SCORE ESTIMATION

A naive way to obtain a confidence score from LLM is P(True) (Kadavath et al., 2022). Given the input-output pair $(x_i, y_i)$, we construct a prompt as $x_i \oplus y_i \oplus I$, where $I$ is a confidence querying prompt, "Is the answer correct? (Yes/No)". The confidence score is defined as the probability of token "**Yes**" in the next position, calculated from the raw logits before any temperature scaling is applied. Due to the KV-cache mechanism (Pope et al., 2022), the additional computational cost is roughly equivalent to generating 10 tokens, which is negligible compared to the typically longer input and output sequences, which is detailed Appendix L. Empirical results suggest that P(True) often lacks calibration, leading to overconfidence in incorrect answers (Tian et al., 2023b). So we aim to

use supervised training to improve the calibration of P(True), helping LLMs produce more reliable confidence scores.

## 3.2 TRAINING DATA GENERATION

Our goal is to create a labeled dataset $D_t = (x, y, c)_i$ without human annotations, where $(x, y)$ is a query–response pair and $c$ is an accurate confidence. To achieve this, we first generate multiple candidate answers for each query and ensure diversity via Dynamic Temperature sampling. Next, we calibrate the confidence of each candidate through Soft Self-Consistency, which integrates the model's intrinsic probability estimate with the overall agreement among different responses.

**Soft Self-Consistency Score.** Previous work has shown that self-consistency scores provide strong zero-shot calibration (Wang et al., 2024a), outperforming P(True) or raw logits as confidence measures (Guo et al., 2017a). To further enhance the reliability of the confidence score in the training set, we introduce a soft self-consistency score, which integrates P(True) with self-consistency and offers a more accurate and robust confidence estimation.

Table 1: Comparison of ECE scores between different confidence estimation methods of Llama-3.1-8B-Instruct on GSM8K and SVAMP.

| Method | GSM8K | SVAMP |
|--------|-------|-------|
| P(True) | 12.03 | 28.94 |
| SC | 4.48 | 4.94 |
| SSC | **3.42** | **3.75** |

For each query $x$, we use the LLM to generate $N$ different responses, each with an associated confidence score. Given the set of triplets $(x, y_n, c_n)$ where $1 \leq n \leq N$, we compute the soft self-consistency (SSC) score as:

$$\text{SSC}(y) = \frac{\sum_{i:y_i=y} c_i}{\sum_{i=1}^{N} c_i}.$$

Using this score, we construct the final training set as $(x, y_i, \text{SSC}(y_i))$, where $\text{SSC}(y_i)$ provides a calibrated confidence estimation for each response.

As shown in Table 1, SSC achieves lower Expected Calibration Error (Guo et al., 2017b) (ECE, definition of which is shown in Appendix H) scores compared to both P(True) and standard self-consistency on GSM8K and SVAMP datasets, demonstrating its effectiveness in providing more reliable confidence estimates.

**Dynamic Temperature.** To generate more diverse and high-quality responses, we adopt the Entropy-based Dynamic Temperature (EDT) Sampling method (Zhang et al., 2024b) when generating each response $y$. By adaptively increasing the temperature when the entropy $H$ of the output distribution is low, EDT promotes greater response diversity while preserving output quality. Formally, the temperature $T(H)$ is defined as:

$$T(H) = \begin{cases} T_0 \times M^{\gamma/H}, & \text{if } T_0 \times M^{\gamma/H} \geq \tau_0, \\ 0, & \text{otherwise}, \end{cases}$$

where $T_0$ is the base temperature, $M$ is a scaling factor, $\gamma$ affects the scale of temperature variations, and $\tau$ is a threshold that sets the temperature to zero if $T_0 \times M^{\gamma/H}$ is below $\tau_0$.

## 3.3 TRAINING OBJECTIVE

We optimize the model's confidence estimation by minimizing the difference between the predicted confidence and the target confidence using the SmoothL1 loss. To ensure that training on confidence estimation does not degrade the model's reasoning ability, we incorporate the standard generation loss of Chain-of-Thought answers into the objective (Huang et al., 2022). Specifically, only responses with confidence scores above a threshold $\eta$ are selected for training to guarantee the quality of the reasoning path. A weighting coefficient $\omega$ is introduced to balance these two loss terms. The overall loss function is formulated as:

$$\mathcal{L}_{\text{total}}(\theta) = \sum_{(x_j, y_j) \in \mathcal{D}} \text{SmoothL1}\Big( p_\theta(\text{Yes} \mid x_j, y_j, I),\ c_j \Big) + \omega \sum_{\substack{(x_i, y_i) \\ c_i > \eta}} \Big( -\log p_\theta\big(y_i \mid x_i\big) \Big).$$

# 4 CALIBRATED TEST-TIME SCALING

In this section, we introduce CaTS, which uses reliable confidence scores from the Self-Calibration trained model. The confidence scores serve as a quality indicator for each response, allowing the model to prioritize more reliable answers. We present confidence-guided variants of three popular test-time scaling approaches: Best-of-N, Self-Consistency, and Adaptive Self-Consistency. Our confidence-guided variants achieve comparable performance with significantly reduced computational costs. Finally, we present the conditions under which this method is effective, along with the theoretical proof.

## 4.1 EARLY STOPPING WITH CALIBRATED CONFIDENCE (CaTS-ES)

CaTS-ES improves the efficiency of Best-of-N by terminating the sampling process once a response with sufficient confidence is found. Given a sequential sampling process where each response $y_i$ is assigned a confidence score $c_i$, we follow this rule:

$$\begin{cases} k \leftarrow k + 1, & \text{if } c_i < \tau, \\ y = y_i, & \text{otherwise.} \end{cases}$$

This means that we continue sampling responses one by one until a response meets the confidence threshold $\tau$, and such a response is selected as the final answer, avoiding unnecessary additional sampling and reducing computational overhead.

## 4.2 SELF-CONSISTENCY WITH CALIBRATED CONFIDENCE (CaTS-SC)

CaTS-SC extends the traditional Self-Consistency approach by incorporating confidence scores into the voting process. Instead of treating all sampled responses equally, we assign each response $y_i$ a confidence score $c_i$, leading to a weighted aggregation:

$$y = \arg\max_z \sum_{i=1}^{N} c_i \, \mathbb{1}(y_i = z).$$

This modification ensures that responses with higher confidence contribute more significantly to the final selection, enhancing robustness by prioritizing more reliable predictions.

## 4.3 ADAPTIVE SELF-CONSISTENCY WITH CALIBRATED CONFIDENCE (CaTS-ASC)

Similar to Self-Consistency with Confidence, we use confidence as the weight when calculating the relative frequency in Adaptive Self-Consistency. CaTS-ASC will follow the rest part in ASC.

$$v_k(z) = \sum_{i=1}^{k} c_i \mathbb{1}(y_i = z), \quad \hat{r}_k(z) = \frac{v_k(z)}{\sum_{i=1}^{k} c_i}.$$

## 4.4 THEORETICAL ANALYSIS

Under the theoretical framework proposed in this work (see Appendix B for a detailed proof), CaTS-SC is guaranteed to exponentially outperform vanilla self-consistency if the confidence signal satisfies the following inequality:

$$\frac{\mu_q^2}{2v_q + \frac{2}{3}\mu_q} > \frac{\mu_{\text{MV}}^2}{2v_{\text{MV}} + \frac{2}{3}\mu_{\text{MV}}}$$

where the terms are defined as:
- $q$ is the **conditional correctness probability**, i.e., the true probability that a model's output is correct given its confidence score. The expectation $\mathbb{E}[\cdot]$ is taken over the distribution of $q$ across all samples.
- $\mu_{\text{MV}}$ is the **mean margin** for vanilla self-consistency (majority voting), representing the average lead of the true answer over a single incorrect competitor, defined as $\mu_{\text{MV}} = \mathbb{E}\left[\frac{kq-1}{k-1}\right]$.

- $v_{\mathrm{MV}}$ is the **margin variance** for vanilla self-consistency (majority voting), representing the uncertainty in the voting process, defined as $v_{\mathrm{MV}} = \mathbb{E}\left[\frac{(1-q)(k^2q+k-2)}{(k-1)^2}\right]$.

- $\mu_q$ is the **mean margin** for CaTS-SC, defined as $\mu_q = \mathbb{E}\left[\frac{q(kq-1)}{k-1}\right]$.

- $v_q$ is the **margin variance** for CaTS-SC, defined as $v_q = \mathbb{E}\left[\frac{q^2(1-q)(k^2q+k-2)}{(k-1)^2}\right]$.

Therefore, our method outperforms the original approach when the provided confidence signal is sufficiently accurate.

## 5 EXPERIMENTS

### 5.1 EXPERIMENT SETUP

**Models.** We conduct experiments on three open-source LLMs: Llama-8B-3.1-Instruct (Dubey et al., 2024), Qwen2.5-7B-Instruct (Team, 2024) and DeepSeek-R1-Distill-Qwen-1.5B (DeepSeek-AI, 2025). These models represent diverse architectures and training strategies, allowing us to test the adaptability of our methods.

**Seed Datasets.** We construct our training dataset with diverse reasoning datasets, including: ARC_easy (Clark et al., 2018), commonsense QA (Talmor et al., 2019), LogiQA (Liu et al., 2020), GSM8K (Cobbe et al., 2021), OpenBookQA (Mihaylov et al., 2018), ReClor (Yu et al., 2020), SciQ (Welbl et al., 2017), SVAMP (Patel et al., 2021) and WinoGrande (Sakaguchi et al., 2019). For each dataset, we randomly sample 2,000 questions (without answers) from the training set to construct our training data. Additional details are shown in Appendix G.

**Evaluation Datasets and Prompts.** We evaluate our methods on three benchmark datasets: ARC-Challenge (Clark et al., 2018), Object-Counting (Suzgun et al., 2022) and MathQA (Amini et al., 2019), covering mathematical and commonsense reasoning tasks in both multiple-choice and open-ended formats. These three datasets are considered out-of-domain as they differ from the datasets used in training, which we refer as in-domain datasets. To evaluate performance in an in-domain setting, we also use the test sets of GSM8K, SVAMP, and ARC_easy. The details of datasets, system prompts and the task prompts of each dataset are shown in Appendix C.

**Baseline Methods.** We include adaptive test-time scaling methods such as Early-Stopping Self-Consistency (ESC) (Li et al., 2024) and Reasoning-Aware Self-Consistency (RASC) (Wan et al., 2024) for comparison. ESC divides the sampling process into sequential windows and halts further sampling when a high-confidence consensus is reached within a window. RASC enhances sampling efficiency by dynamically evaluating both the generated answers and their reasoning paths. To ensure a fair comparison, we reproduce these two methods on our Self-Calibration trained models. We further introduce an untrained confidence-informed self-consistency(CISC) (Taubenfeld et al., 2025) as a baseline with probability as confidence and Self-Certainty (Kang et al., 2025) which uses confidence for ranking.

**Experiment Setting.** To ensure a fair comparison across different test-time scaling methods, we use the same sample budgets for each of them. Sample budget refers to the average number of responses each method samples per query. For dynamic methods such as CaTS-ES and CaTS-ASC, we calibrate their threshold for each dataset so that the actual number of samples collected in practice closely matches, but slightly under, the target budget.

### 5.2 RESULTS

Table 2 shows the accuracy comparison of different methods with a sample budget of 16. We observe that CaTS-SC, CaTS-ES, and CaTS-ASC consistently outperform their base counterparts. On Llama-3.1-8B-Instruct, CaTS-SC surpasses SC on MathQA (73.7 to 83.6), while on DeepSeek-R1-Distill-1.5B, CaTS-ES outperforms Best-of-N on ARC_challenge (54.1 to 66.5). These results highlight that integrating calibrated confidence enhances test-time scaling with the same sampling budget. We also provide evaluation results on reasoning datasets such as MMLU_pro (Wang et al., 2024c), Hellaswag (Zellers et al., 2019), and GPQA (Rein et al., 2024) in Table 10.

Table 2: Accuracy comparison of different test-time scaling methods across three language models when the sample budget equals 16. The evaluation is conducted on three datasets: Obj_C. (Object_Counting), MathQA, and ARC_C. (ARC_Challenge). "Sample budget" refers to the average number of responses sampled per query. The improvements of CaTS over their baselines are shown in parentheses. Results for sample budget set to 4 are shown in Appendix D.

| Methods | Llama-3.1-8B-Instruct | | | Qwen2.5-7B-Instruct | | | DeepSeek-R1-Distill-1.5B | | |
|---|---|---|---|---|---|---|---|---|---|
| | Obj_C. | MathQA | ARC_C. | Obj_C. | MathQA | ARC_C. | Obj_C. | MathQA | ARC_C. |
| SC | 69.1 | 73.7 | 85.2 | 76.8 | 83.3 | **90.1** | 64.9 | 89.4 | 60.8 |
| CISC | 72.8 | 81.2 | 86.4 | 80.2 | 81.1 | 89.2 | 66.4 | 90.8 | 61.7 |
| Self-Certainty | 73.2 | 81.5 | 86.8 | 80.6 | 82.4 | 89.5 | 66.9 | 90.7 | 62.3 |
| CaTS-SC | **76.8** (+7.7) | **83.6** (+9.9) | **87.7** (+2.5) | 81.2 (+4.4) | **87.8** (+4.5) | 90.8 (+0.7) | **70.8** (+5.9) | **91.8** (+2.4) | **66.5** (+5.7) |
| Best-of-N | 62.3 | 73.7 | 84.5 | 72.4 | 83.8 | 91.1 | 48.1 | 87.8 | 54.1 |
| CaTS-ES | **76.8** (+14.5) | **83.6** (+9.9) | **87.7** (+3.2) | 81.2 (+8.8) | **87.8** (+4.0) | 90.8 (-0.3) | **70.8** (+22.7) | 91.6 (+3.8) | **66.5** (+12.4) |
| ASC | 67.9 | 72.7 | 84.6 | 77.2 | 83.2 | 90.9 | 64.5 | 89.4 | 59.5 |
| CaTS-ASC | 75.2 (+7.3) | 81.9 (+9.2) | 86.6 (+2.0) | **81.6** (+4.4) | 87.2 (+4.0) | **91.2** (+0.3) | 70.4 (+5.8) | **91.8** (+2.4) | 65.1 (+5.6) |
| ESC | 76.0 | 81.0 | 87.1 | 81.2 | 86.3 | 91.0 | **70.8** | 91.3 | 65.6 |
| RASC | 76.0 | 81.4 | 87.3 | 81.2 | 86.4 | 90.3 | 70.8 | 91.4 | 65.8 |

# 6 ANALYSIS

## 6.1 PERFORMANCE COMPARISON UNDER DIFFERENT SAMPLE BUDGETS

Increasing the sample budget allows for selecting higher-quality outputs but comes at the cost of greater computational expense. To evaluate this trade-off, we compare different methods across multiple sample budgets and visualize their performance trends. All methods use the same responses generated by Self-Calibration trained models. As shown in Figure 3, all methods achieve better accuracy as the number of responses increases. Our confidence-guided approaches consistently outperform their original counterparts in most settings. When the sample budget is small, Best-of-N performs better than CaTS-ES because early stopping might stop too soon with a low threshold, missing a better response. The results of other models and datasets are shown in Appendix F.

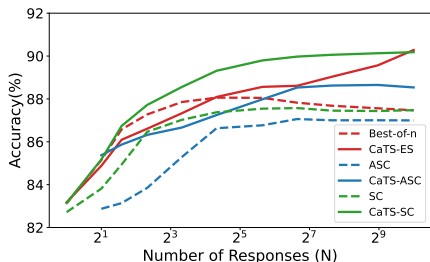

Figure 3: Accuracy over varying sample budgets of different inference strategies on MathQA using self-calibrated Qwen-2.5-7B-Instruction.

## 6.2 CALIBRATED CONFIDENCE SCORE COMPARED TO REWARD SCORE FROM REWARD MODELS

We compare our self-calibrated confidence scores with established open-source reward model approaches. A reward model is an additional scoring model used to evaluate the quality of generated responses (Christiano et al., 2017). Deployment of a reward model can introduce several limitations: (1) Reward scores are often unbounded or require dataset-specific normalization, thus difficult to apply a universal threshold for filtering or reweighting responses; (2) Running an extra reward model increases inference time;

Table 3: Accuracy of Best-of-16 on two models (Llama-3.1-8B-Instruct and Qwen-2.5-7B-Instruct) on three datasets between self-calibrated confidence scores and reward scores from additional reward models.

| Model | Dataset | Reward | Confidence |
|---|---|---|---|
| Llama | MathQA | 82.1 | **84.0** |
| | Object Counting | **72.6** | 72.0 |
| | ARC_Challenge | 86.2 | **86.6** |
| Qwen | MathQA | **87.5** | 86.8 |
| | Object Counting | **76.6** | 76.4 |
| | ARC_Challenge | 89.6 | **89.8** |

and (3) A dedicated reward model requires additional GPU memory, and is less efficient for large-scale deployment.

Table 4: Self-Calibration results across both in-domain and out-of domain datasets on three different models. We use Self-Cal. to denote Self-Calibration in the table.

| Dataset | Metric | Llama-3.1-8B-Instruct | | Qwen2.5-7B-Instruct | | DS-R1-Distill-1.5B | |
|---------|--------|---------|-----------|---------|-----------|---------|-----------|
| | | Vanilla | Self-Cal. | Vanilla | Self-Cal. | Vanilla | Self-Cal. |
| *In-Domain Datasets* | | | | | | | |
| GSM8K | ECE ↓ | 13.70 | **3.79** | 87.39 | **16.88** | 46.66 | **40.03** |
| | AUC ↑ | 72.43 | **75.36** | 68.61 | **82.21** | **64.31** | 55.57 |
| | ACC ↑ | 77.44 | **80.43** | **89.41** | 88.74 | 73.38 | **75.36** |
| SVAMP | ECE ↓ | 28.03 | **10.17** | 89.60 | **24.49** | 30.40 | **12.00** |
| | AUC ↑ | 74.17 | **75.79** | 75.10 | **87.46** | 49.33 | **71.27** |
| | ACC ↑ | 72.60 | **75.29** | 90.48 | **92.00** | 52.27 | **57.48** |
| ARC_easy | ECE ↓ | 5.45 | **5.00** | 57.58 | **5.62** | 20.19 | **11.36** |
| | AUC ↑ | **81.16** | 76.89 | 66.10 | **76.75** | 62.89 | **66.86** |
| | ACC ↑ | 87.73 | **89.21** | **92.11** | 92.01 | 54.00 | **56.74** |
| *Out-of-Domain Datasets* | | | | | | | |
| ARC_challenge | ECE ↓ | 7.01 | **6.03** | 55.19 | **10.11** | 11.42 | **5.46** |
| | AUC ↑ | **80.67** | 80.41 | 64.21 | **78.33** | 64.07 | **65.27** |
| | ACC ↑ | 80.87 | **82.38** | **89.37** | 89.05 | 43.39 | **45.77** |
| Object Counting | ECE ↓ | 27.85 | **9.69** | 72.41 | **5.82** | 47.26 | **4.60** |
| | AUC ↑ | 53.84 | **59.47** | 68.07 | **81.02** | 50.39 | **67.61** |
| | ACC ↑ | 60.68 | **65.88** | 72.41 | **74.22** | 55.33 | **58.13** |
| MathQA | ECE ↓ | 12.55 | **8.64** | 62.35 | **18.92** | 13.16 | **4.34** |
| | AUC ↑ | 85.23 | **87.21** | **72.48** | 69.80 | **78.89** | 66.09 |
| | ACC ↑ | 44.18 | **52.34** | 49.85 | **64.18** | 37.69 | **43.21** |

For our analysis, we use the following reward models for comparison: for Llama-3.1-Instruct, we use the reward model from RLHFlow [1] (Dong et al., 2024); for Qwen-2.5, we utilize its official Process Reward Model (PRM) [2] (Zhang et al., 2025). For PRM, we use the lowest reward score across all steps. We ensure the size of each reward model matches with their corresponding base models.

Table 3 shows that our self-calibrated confidence scores achieve similar performance to reward model scores across all datasets when using Best-of-N. This means that our method, by generating approximately 10 additional tokens, achieves a performance comparable to that of an extra reward model of the same size. We further conduct a time analysis in Appendix J to demonstrate the efficiency of our approach.

## 6.3 EVALUATION ON SELF-CALIBRATION

To prove the effectiveness of our Self-Calibration model training to output accurate confidence estimations, we adopt three standard metrics for evaluating model calibration: ECE, Area Under the Receiver Operating Characteristic Curve (AUC) (Hendrycks & Gimpel, 2017), and accuracy (ACC). The explanation and details of these metrics can be found in Appendix H.

**Results.** In Table 4, we compare our models trained on Self-Calibration objective with their vanilla base models on multiple in-domain and out-of-domain datasets. Self-Calibration trained models consistently lower the ECE score while generally improving accuracy. On GSM8K, Self-Calibration reduces ECE from 13.70 to 3.79 while improving accuracy from 77.44% to 80.43%. Even in cases where ECE slightly increases, such as ARC_easy for Llama-3.1-8B-Instruct, accuracy still improves from 87.73% to 89.21%. Moreover, the strong results on out-of-

Table 5: Ablation study results on MathQA and Object Counting in Llama-3.1-8B-Instruct. "w/o L1-smooth" means using MSE loss instead of L1-smooth.

| Method | MathQA | | Object Counting | |
|--------|--------|--------|--------|--------|
| | ECE ↓ | ACC ↑ | ECE ↓ | ACC ↑ |
| ours (full) | 8.64 | 52.34 | 9.69 | 65.88 |
| w/o EDT | 9.14 | 51.44 | 10.40 | 62.88 |
| w/o SSC | 10.85 | 52.18 | 16.02 | 61.12 |
| w/o L1-smooth | 6.43 | 50.86 | 10.48 | 56.48 |

domain tasks demonstrate the generalizability of our method, as seen in MathQA, where accuracy improves from 49.85% to 64.18% for Qwen2.5-7B-Instruct.

[1] https://huggingface.co/RLHFlow/Llama3.1-8B-ORM-Mistral-Data
[2] https://huggingface.co/Qwen/Qwen2.5-Math-PRM-7B

**Ablation Study.** We conduct an ablation study to investigate the impact of key components in Self-Calibration, including Dynamic Temperature (EDT), Soft Self-Consistency (SSC), and L1-smooth loss. Table 5 presents our ablation results on the MathQA and Obj_Cou datasets.

Removing the dynamic temperature or the soft self-consistency score leads to noticeable increases in ECE and/or drops in accuracy. Meanwhile, replacing the L1-smooth objective with MSE achieves slightly lower ECE on MathQA but reduces accuracy on both tasks, suggesting that our chosen loss formulation is more robust overall. These results demonstrate that each module contributes to model calibration and reasoning performance.

## 6.4 CAN OTHER CONFIDENCE QUERYING PROMPTS WORK WELL?

Since our confidence-based approach was trained using a specific confidence querying prompt, we explore whether alternative prompts can achieve similar performance during inference. This analysis is crucial for understanding the robustness of confidence querying prompts different from the training prompt.

We evaluate 6 alternative prompts (listed in Appendix E.1) at inference time. Table 6 shows that despite training with a specific prompt, other prompts yield comparable performance, with only minor variations. This suggests that our confidence querying approach is robust to prompt changes and our training framework improves model calibration rather than overfitting to a specific prompt.

Table 6: Accuracy comparison between the original prompt and 6 alternative querying prompts on self-calibrated Llama-3.1-8B-Instruct. Results are reported as $\text{mean}_{\pm\text{std}}$.

| Dataset | Method | Original | New |
|---------|--------|----------|-----|
| MathQA | CaTS-ES | 81.7 | $81.52_{\pm 0.30}$ |
|  | CaTS-ASC | 81.9 | $81.80_{\pm 0.21}$ |
|  | CaTS-SC | 82.1 | $81.63_{\pm 0.20}$ |
| Obj_C. | CaTS-ES | 67.2 | $70.80_{\pm 1.99}$ |
|  | CaTS-ASC | 74.8 | $74.07_{\pm 1.03}$ |
|  | CaTS-SC | 74.4 | $73.40_{\pm 0.75}$ |
| ARC_C. | CaTS-ES | 86.2 | $86.62_{\pm 0.20}$ |
|  | CaTS-ASC | 86.6 | $86.63_{\pm 0.05}$ |
|  | CaTS-SC | 86.4 | $86.35_{\pm 0.25}$ |

# 7 RELATED WORK

## 7.1 TEST-TIME SCALING

Snell et al. (2024b); Chen et al. (2025b) studied optimal test-time compute allocation to significantly enhance efficiency. Self-Enhanced tree search frameworks (Bi et al., 2024; Lample et al., 2022; Koh et al., 2024; Zheng et al., 2025) aggregate multiple reasoning paths and employ sparse activation strategies. Beyond that, step-wise verifiers are leveraged to dynamically prune the search tree (Wang et al., 2022a; Li et al., 2022; Lightman et al., 2023a). Combining different versions of the same query can also improve the performance (Huang et al., 2024). Scaling (Chen et al., 2025a; Welleck et al., 2022; Wang et al., 2024b; Chen et al., 2023; Madaan et al., 2023; Aggarwal et al., 2024) that iteratively refines model outputs to improve performance in complex tasks.

## 7.2 MODEL CALIBRATION

Model calibration aims to align a model's confidence with its accuracy. Prior research has explored scaling-based methods (Deng et al., 2023; Guo et al., 2017b; Zhang et al., 2020) and nonparametric techniques like binning (Zadrozny & Elkan, 2001). More recent work has introduced verbalized confidence, prompting models to directly output confidence scores (Lin et al., 2022). Most studies focus on pre-trained and instruction-tuned LLMs (Lin et al., 2022; Han et al., 2024), others investigate RLHF-trained LLMs and propose calibration through prompting strategies (Xiong et al., 2023; Tian et al., 2023b).

## 7.3 LLM VERIFIER

Our approach is closely related to LLM-based verifiers (Zhao et al., 2025), as both aim to evaluate whether a generated response meets correctness criteria. Lightman et al. (2023b) trained verifiers that assess the correctness of generated solutions, enhancing the selection of accurate responses. Zhang et al. (2024a) trained verifiers using next-token prediction to enhance reasoning performance in large language models. GenRM (Mahan et al., 2024) is an iterative algorithm that trains LLMs on self-generated reasoning traces to align synthetic preference labels.

## 8    CONCLUSION

We improve the efficiency of test-time scaling methods in LLMs with reliable confidence estimation. Our Self-Calibration enhances LLM confidence estimation in one forward pass, without requiring any labeled data. We then propose CaTS by dynamically adjusting sampling strategies based on calibrated confidence scores. Experiments show that our approaches consistently outperform baselines under the same sample budget. Our findings suggest that reliable confidence estimation and dynamic sampling can substantially enhance the effectiveness and efficiency of repeated sampling.

## REPRODUCIBILITY STATEMENT

To help reproduce our work, we present the following:

- **Datasets and Models:** All datasets and models used in our experiments are publicly available.
- **Prompts:** The specific prompts used to in our experiments are detailed in Appendix C.
- **Hyperparameters:** A full list of hyperparameters for all experiments is provided in Appendix G.

## ACKNOWLEDGMENTS

We thank the anonymous reviewers and the area chair for their time, effort, and constructive suggestions, which have greatly improved the quality of this paper. We thank Haolin Liu (University of Virginia) for helpful discussions regarding the theoretical part of this paper. This research was supported in part by the NVIDIA Academic Grant Program and WashU Ignite Interdisciplinary Grants.

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

## A  THE USE OF LARGE LANGUAGE MODELS (LLMS)

We acknowledge the use of large language models (LLMs) as assistive tools in this research, with usage limited to refining grammar and improving language clarity in the manuscript and writing utility scripts for data preprocessing and postprocessing; all outputs from these models were meticulously reviewed, revised, and verified by the authors, who retain full responsibility for all content presented in this paper.

# B  THEORETICAL FRAMEWORK AND PROOFS

## B.1  SETTING AND ASSUMPTIONS

Let $\mathcal{A} = \{1, \ldots, K\}$, $K \geq 2$, be the set of possible answers and let the (unique) ground-truth answer be $T \in \mathcal{A}$. From an LLM we draw $n$ samples

$$(X_i, C_i) \in \mathcal{A} \times [0, 1] \qquad (i = 1, \ldots, n),$$

where $X_i$ is the answer of sample $i$ and $C_i$ is a scalar confidence score of sample $i$.

**Assumption 1** (Conditional i.i.d.). *Conditional on $T$, the pairs $\{(X_i, C_i)\}_{i=1}^n$ are independent and identically distributed.*

**Assumption 2** (Monotone Confidence). *There exists a (measurable) function $q : [0, 1] \to (\frac{1}{K}, 1)$, strictly increasing, such that for any $t \in \mathcal{A}$ and any $c \in [0, 1]$,*

$$\mathbb{P}(X_i = t \mid C_i = c, \, T = t) \;=\; q(c).$$

*We refer to $q(c)$ as the* conditional correctness probability at confidence $c$.

**Assumption 3** (Distribution of wrong labels). *When $X_i \neq T$, the allocation of mass among wrong labels does not depend on the particular true class. A canonical and widely used choice is the symmetric model:*

$$\mathbb{P}(X_i = b \mid C_i = c, \, T = t) \;=\; \frac{1 - q(c)}{K - 1} \quad \text{for each } b \neq t.$$

**Definition 1** (Weighted voting rules). *Given a nonnegative weight function $h : [0, 1] \to \mathbb{R}_+$, define the per-class score and decision*

$$S_a^{(h)} \;:=\; \sum_{i=1}^n \mathbf{1}\{X_i = a\}\, h\big(q(C_i)\big), \qquad \widehat{T}_h \;:=\; \arg\max_{a \in \mathcal{A}} S_a^{(h)}.$$

*Two special cases:*

- *Majority Voting (MV): $h \equiv 1$.*

- *$q$-**weighted** voting (CWMV-q): $h(q) = q$.*

*When comparing a true class $a$ to a competitor $b \neq a$, define the pairwise* margin summand

$$Y_i^{(h)} \;:=\; \big(\mathbf{1}\{X_i = a\} - \mathbf{1}\{X_i = b\}\big)\, h\big(q(C_i)\big), \qquad \text{so that} \quad S_a^{(h)} - S_b^{(h)} = \sum_{i=1}^n Y_i^{(h)}.$$

## B.2  MOMENTS OF THE PAIRWISE MARGIN

The next lemma gives exact conditional moments used throughout.

**Lemma 2** (Mean and variance of the pairwise summand). *Fix a true class $a$ and a competitor $b \neq a$. Under Assumptions 1–3, for any weight $h$ with essential supremum $H := \|h\|_\infty < \infty$ and for $q_i := q(C_i)$,*

$$\mathbb{E}\Big[Y_i^{(h)} \,\big|\, q_i, \, T = a\Big] = \frac{Kq_i - 1}{K - 1}\, h(q_i),$$

$$\mathrm{Var}\Big(Y_i^{(h)} \,\big|\, q_i, \, T = a\Big) = \frac{(1 - q_i)\,(K^2 q_i + K - 2)}{(K - 1)^2}\, h(q_i)^2,$$

$$\text{and} \quad |Y_i^{(h)}| \leq H \quad a.s.$$

*Consequently, with $q \overset{d}{=} q(C_1)$,*

$$\mu_h \;:=\; \mathbb{E}\Big[Y_i^{(h)} \mid T = a\Big] = \mathbb{E}\left[\frac{Kq - 1}{K - 1}\, h(q)\right],$$

$$v_h \;:=\; \mathbb{E}\Big[\mathrm{Var}\Big(Y_i^{(h)} \mid q, \, T = a\Big)\Big] = \mathbb{E}\left[\frac{(1 - q)\,(K^2 q + K - 2)}{(K - 1)^2}\, h(q)^2\right].$$

*Proof.* Condition on $q_i = c$. Under $T = a$ we have by Assumptions 2–3:

$$\mathbb{P}(X_i = a \mid c, T = a) = q(c), \quad \mathbb{P}(X_i = b \mid c, T = a) = \frac{1 - q(c)}{K - 1}$$

$$\mathbb{P}(X_i \notin \{a, b\} \mid c, T = a) = \frac{(K - 2)(1 - q(c))}{K - 1}.$$

Thus $Y_i^{(h)}$ takes values $+h(q(c))$, $-h(q(c))$, and $0$ with those probabilities, and

$$\mathbb{E}[Y_i^{(h)} \mid c, T = a] = h(q(c))\Big(q(c) - \frac{1 - q(c)}{K - 1}\Big) = \frac{Kq(c) - 1}{K - 1}\, h\big(q(c)\big).$$

For the variance, $E\Big[(Y_i^{(h)})^2 \mid c, T = a\Big] = h(q(c))^2 \Big(q(c) + \frac{1 - q(c)}{K - 1}\Big)$. Subtracting the squared mean yields

$$\mathrm{Var}(Y_i^{(h)} \mid c, T = a) = h(q(c))^2 \Big(q(c) + \frac{1-q(c)}{K-1} - \frac{(Kq(c)-1)^2}{(K-1)^2}\Big) = \frac{(1-q(c))(K^2 q(c)+K-2)}{(K-1)^2}\, h(q(c))^2,$$

where the last equality is an algebraic identity. Boundedness is immediate since $|Y_i^{(h)}| \leq h(q_i) \leq H$. Taking outer expectations in $q$ gives the stated $\mu_h$ and $v_h$. $\qquad\square$

**Positivity of the mean.** Since $q > \frac{1}{K}$ almost surely by Assumption 2 and $h \geq 0$ is not almost surely zero, we have $\mu_h > 0$.

### B.3 CONSISTENCY AND NON-ASYMPTOTIC EXPONENTIAL BOUNDS

**Theorem 3** (Strong consistency and exponential tails). *Assume 1–3 and let $H = \|h\|_\infty < \infty$. For any true class $a$ and any competitor $b \neq a$,*

$$\mathbb{P}\Big(S_b^{(h)} \geq S_a^{(h)} \mid T = a\Big) = \mathbb{P}\bigg(\sum_{i=1}^n Y_i^{(h)} \leq 0 \,\Big|\, T = a\bigg)$$

$$\leq \exp\bigg(-\frac{n\,\mu_h^2}{2\,v_h + \frac{2}{3} H\,\mu_h}\bigg) \leq \exp\bigg(-\frac{n\,\mu_h^2}{2\,v_h + \frac{2}{3} H^2}\bigg), \qquad (1)$$

*where $\mu_h$ and $v_h$ are given in Lemma 2. Consequently,*

$$\mathbb{P}\Big(\widehat{T}_h \neq a \mid T = a\Big) \leq (K - 1)\exp\bigg(-\frac{n\,\mu_h^2}{2\,v_h + \frac{2}{3} H\,\mu_h}\bigg), \qquad (2)$$

*and in particular $\mathbb{P}(\widehat{T}_h \neq T) \to 0$ exponentially fast as $n \to \infty$ (strong consistency).*

*Proof.* By Definition 1, $S_a^{(h)} - S_b^{(h)} = \sum_{i=1}^n Y_i^{(h)}$ where $\{Y_i^{(h)}\}$ are independent given $T = a$. Each summand is bounded by $|Y_i^{(h)}| \leq H$ and has positive mean $\mu_h = \mathbb{E}[Y_i^{(h)} \mid T = a] > 0$ and average conditional variance $v_h$ from Lemma 2. Apply Bernstein's inequality for bounded independent variables to the centered sum $\sum_i (Y_i^{(h)} - \mu_h)$:

$$\mathbb{P}\bigg(\sum_{i=1}^n Y_i^{(h)} - n\mu_h \leq -n\mu_h\bigg) \leq \exp\bigg(-\frac{n\mu_h^2}{2v_h + \frac{2}{3} H\mu_h}\bigg),$$

which is equation 1. The second inequality follows from $\mu_h \leq H$. A union bound over $b \in \mathcal{A} \setminus \{a\}$ gives equation 2. Exponential decay in $n$ is immediate. $\qquad\square$

### B.4 WHEN $q$-WEIGHTING IS EXPONENTIALLY BETTER THAN MV

We now compare the *error exponents* (the constants in the exponential upper bounds) of MV and $q$-weighting under the same assumptions.

**Definition 4** (Bernstein error exponent). *For a weight $h$ with $H = \|h\|_\infty$, define the per-pair Bernstein exponent*

$$\mathcal{E}^{(a \succ b)}(h) := \frac{\mu_h^2}{2v_h + \frac{2}{3} H\,\mu_h},$$

*with $\mu_h$ and $v_h$ as in Lemma 2. The overall top-1 error satisfies $\mathbb{P}(\widehat{T}_h \neq a \mid T = a) \leq (K - 1)\exp(-\mathcal{E}^{(a \succ b)}(h)\, n)$ for each fixed $b \neq a$ (Theorem 3).*

**Theorem 5** (Explicit criterion for exponential advantage). *Under Assumptions 1–3, with MV ($h \equiv 1$) and $q$-weighting ($h(q) = q$) and $H = 1$, define*

$$\mu_{\mathrm{MV}} = \mathbb{E}\left[\frac{Kq - 1}{K - 1}\right], \qquad v_{\mathrm{MV}} = \mathbb{E}\left[\frac{(1 - q)(K^2 q + K - 2)}{(K - 1)^2}\right],$$
$$\mu_q = \mathbb{E}\left[\frac{q(Kq - 1)}{K - 1}\right], \qquad v_q = \mathbb{E}\left[\frac{q^2(1 - q)(K^2 q + K - 2)}{(K - 1)^2}\right].$$

*If the inequality*

$$\frac{\mu_q^2}{2v_q + \frac{2}{3}\mu_q} \; > \; \frac{\mu_{\mathrm{MV}}^2}{2v_{\mathrm{MV}} + \frac{2}{3}\mu_{\mathrm{MV}}} \tag{3}$$

*holds (for the distribution of $q(C)$ induced by $C$), then for each competitor $b \neq a$,*

$$\mathcal{E}^{(a \succ b)}(q) \; > \; \mathcal{E}^{(a \succ b)}(\mathrm{MV}) \quad \Longrightarrow \quad \mathbb{P}(\widehat{T}_q \neq a \mid T = a) \; \leq \; (K - 1)\, e^{-\mathcal{E}^{(a \succ b)}(q)\, n} \; \ll \; (K - 1)\, e^{-\mathcal{E}^{(a \succ b)}(\mathrm{MV})\, n}.$$

*In words: $q$-weighting has a strictly larger error exponent and thus* exponentially dominates *majority voting.*

*Proof.* This is an immediate comparison of the Bernstein exponents from Theorem 3. Under the stated assumption equation 3, we have $\mathcal{E}^{(a \succ b)}(q) > \mathcal{E}^{(a \succ b)}(\mathrm{MV})$ for every pair $(a, b)$, yielding the claimed dominance pairwise and hence for the top-1 error by the union bound. □

**Readable sufficient condition.** The technical inequality equation 3 is *verifiable from held-out data*. One can also give transparent sufficient conditions ensuring it holds. A useful example is a *two-tier confidence mixture*:

**Lemma 6** (Two-tier sufficient condition). *Suppose $q(C)$ takes values $q_{\mathrm{hi}} \in (\frac{1}{K}, 1)$ with probability $\pi \in (0, 1)$ and $q_{\mathrm{lo}} \in [\frac{1}{K}, q_{\mathrm{hi}})$ with probability $1 - \pi$. Then the inequality equation 3 holds whenever*

$$\frac{\left(\pi q_{\mathrm{hi}}(K q_{\mathrm{hi}} - 1) + (1 - \pi) q_{\mathrm{lo}}(K q_{\mathrm{lo}} - 1)\right)^2}{\pi q_{\mathrm{hi}}^2(1 - q_{\mathrm{hi}})(K^2 q_{\mathrm{hi}} + K - 2) + (1 - \pi) q_{\mathrm{lo}}^2(1 - q_{\mathrm{lo}})(K^2 q_{\mathrm{lo}} + K - 2)}$$
$$+ \frac{2}{3}(K - 1)\left(\pi q_{\mathrm{hi}}(K q_{\mathrm{hi}} - 1) + (1 - \pi) q_{\mathrm{lo}}(K q_{\mathrm{lo}} - 1)\right)$$
$$> \frac{\left(\pi(K q_{\mathrm{hi}} - 1) + (1 - \pi)(K q_{\mathrm{lo}} - 1)\right)^2}{\pi(1 - q_{\mathrm{hi}})(K^2 q_{\mathrm{hi}} + K - 2) + (1 - \pi)(1 - q_{\mathrm{lo}})(K^2 q_{\mathrm{lo}} + K - 2)}$$
$$+ \frac{2}{3}(K - 1)\left(\pi(K q_{\mathrm{hi}} - 1) + (1 - \pi)(K q_{\mathrm{lo}} - 1)\right).$$

*In particular, if $\pi$ and $(q_{\mathrm{hi}} - q_{\mathrm{lo}})$ are not too small (i.e., there is a nontrivial mass of genuinely high-confidence votes), the inequality is satisfied; thus CWMV-$q$ exponentially dominates MV.*

*Proof.* Substitute the two-point distribution into the definitions of $\mu_q, v_q, \mu_{\mathrm{MV}}, v_{\mathrm{MV}}$ and check equation 3. The final statement follows by continuity: as $\pi(q_{\mathrm{hi}} - q_{\mathrm{lo}})$ grows, the left-hand side increases while the right-hand side decreases, because $q$-weighting shrinks the contribution of low-confidence votes (reducing the denominator via the $q^2$ factor) while preserving the positive drift on high-confidence votes (numerator). □

**Binary simplification ($K = 2$).** For $K = 2$ the formulas simplify to

$$\mu_{\mathrm{MV}} = \mathbb{E}[\, 2q - 1 \,], \quad v_{\mathrm{MV}} = \mathbb{E}[\, 4q(1 - q) \,], \qquad \mu_q = \mathbb{E}[\, q(2q - 1) \,], \quad v_q = \mathbb{E}[\, 4q^3(1 - q) \,],$$

and equation 3 becomes

$$\frac{\left(\mathbb{E}[q(2q - 1)]\right)^2}{2\, \mathbb{E}[4q^3(1 - q)] + \frac{2}{3}\mathbb{E}[q(2q - 1)]} \; > \; \frac{\left(\mathbb{E}[2q - 1]\right)^2}{2\, \mathbb{E}[4q(1 - q)] + \frac{2}{3}\mathbb{E}[2q - 1]}.$$

# C PROMPTS

## C.1 DETAILS ABOUT TEST DATASET

ARC-Challenge includes difficult science questions requiring external knowledge and reasoning. Object-Counting focuses on numerical and spatial reasoning by counting objects. MathQA tests mathematical problem-solving across arithmetic, algebra, and calculus.

## C.2 SYSTEM PROMPT

Here we show the system prompt to let the model generate responses for Chain-of-Thoughts and format for extracting the final results.

```
For the following question, provide a step-by-step
explanation of your thought process.
Use the format demonstrated below for your response.

'''Example Format:
Explanation: <Your detailed explanation here,
outlining how you arrived at your answer.>
Answer: <Insert your concise answer here, which
should include a {answer_type} (e.g., {demo})>

Ensure that your response strictly adheres to this
format. Explicitly include the words 'Explanation:',
'Answer:'.
```

The answer type includes "option letter" and "number".

## C.3 DATASET PROMPTS

We show the prompts for each dataset in Table 7. All datasets and models are open-sourced.

Table 7: Query templates for each dataset .

| Dataset | Query Template |
|---|---|
| gsm8k | Question: {question}\n |
| sciq | Question: {question}\nOptions:\n{options_text}\n |
| commonsense_qa | Question: {question}\nOptions:\n{options_text}\n |
| winogrande | Question: {sentence}\nOptions:\nA. {option1}\nB. {option2}\n |
| openbookqa | Question: {question}\nOptions:\n {options_text}\n |
| reclor | Passage:\n{passage}\n\nQuestion: {question}\n\n Options:\n{options_text}\n |
| math_qa | Problem: {problem_text}\nOptions:\n{options_block}\n |
| arc_challenge | Question: {question}\n Options:\n{options_str}\n |
| arc_easy | Question: {question}\nOptions:\n{options_str}\n |
| logiqa | Article:\n{context}\n\nQuestion: {question}\n\n Options:\n{options_text}\n |
| svamp | Question: {Body + Question}\n |
| gpqa | {Question}\nOptions:\n{options_text}\n |
| aqua_rat | Question: {question}\nOptions:\n{options_text}\n |

# D FULL MAIN RESULTS

Here we show the main results when sample budget = 4 in Table 8.

When the sample budget is small, the model has limited opportunities to explore different reasoning paths. In this scenario, output variability is often high, and having an additional confidence signal (as

Table 8: Accuracy comparison of different test-time scaling methods across three language models. The evaluation is conducted on three datasets: Obj_C. (Object_Counting), MathQA, and ARC_C. (ARC_Challenge). "Sample budget" refers to the average number of responses sampled per query. The improvements of confidence-augmented methods over their baselines are shown in parentheses. All methods use the same responses generated by Self-Calibration trained models.

| Methods | Llama-3.1-8B-Instruct | | | Qwen2.5-7B-Instruct | | | DeepSeek-R1-Distill-1.5B | | |
| --- | --- | --- | --- | --- | --- | --- | --- | --- | --- |
| | Obj_C. | MathQA | ARC_C. | Obj_C. | MathQA | ARC_C. | Obj_C. | MathQA | ARC_C. |
| *Sample budget = 4* | | | | | | | | | |
| SC | 65.1 | 71.5 | 83.8 | 74.4 | 82.7 | 89.9 | 57.7 | 79.2 | 48.2 |
| CaTS-SC | **72.8** (+7.7) | **82.1** (+10.6) | **86.4** (+2.6) | **78.4** (+4.0) | **86.9** (+4.2) | **90.3** (+0.4) | **64.0** (+6.3) | **91.2** (+12.0) | **63.2** (+15.0) |
| Best-of-N | 60.7 | 73.5 | 84.5 | 72.0 | 83.4 | 89.7 | 50.1 | 77.8 | 44.2 |
| CaTS-ES | **67.2** (+6.5) | **81.7** (+8.2) | **86.2** (+1.7) | **78.4** (+6.4) | **87.1** (+3.7) | **90.1** (+0.4) | **56.0** (+5.9) | **90.6** (+12.8) | **59.0** (+14.8) |
| ASC | 67.5 | 72.7 | 84.6 | 75.2 | 83.2 | 88.9 | 55.3 | 79.1 | 48.5 |
| CaTS-ASC | **74.8** (+7.3) | **81.9** (+9.2) | **86.6** (+2.0) | **80.0** (+4.8) | **87.2** (+4.0) | **90.6** (+1.7) | **62.8** (+7.5) | **91.6** (+12.5) | **64.6** (+16.1) |
| ESC | 72.0 | 78.6 | 85.8 | **80.0** | 86.9 | 89.6 | 58.0 | 91.2 | 63.0 |
| RASC | 72.4 | 79.0 | 85.8 | **80.0** | 86.4 | 89.8 | 62.6 | 91.2 | 63.1 |

in ASC w/ Conf.) is essential for filtering out noisy or incorrect responses. This confidence-augmented method helps select the most promising candidate under tight sampling constraints.

However, when the sample budget increases, the model can generate more candidate solutions, which typically raises the chance of hitting the correct answer. In this setting, Early Stopping approach—especially when coupled with a high confidence threshold—can terminate as soon as it encounters a correct reasoning path.

# E  FULL RESULTS OF DIFFERENT CONFIDENCE QUERYING PROMPTS

## E.1  CONFIDENCE QUERYING PROMPTS

We show the 6 confidence querying prompts we used in Sec. 6.4.

- $I_1$: Is this the correct answer?
- $I_2$: Does this answer seem right?
- $I_3$: Is this the right answer?
- $I_4$: Is the given answer accurate?
- $I_5$: Would you say this answer is correct?
- $I_6$: Is this response correct?

## E.2  RESULTS OF DIFFERENT QUERYING PROMPTS

In Table 9, we show the results of different confidence querying prompts for tuned LLama-3.1-8B-Instruct.

Table 9: The results for different confidence querying prompt.

| Dataset | Method | 1 | 2 | 3 | 4 | 5 | 6 | Original |
| --- | --- | --- | --- | --- | --- | --- | --- | --- |
| **MathQA** | CaTS-ES | 81.7 | 81.4 | 81.7 | 81.3 | 81.1 | **81.9** | 81.7 |
| | CaTS-ASC | 81.9 | 81.9 | 81.8 | 81.8 | 81.4 | **82.0** | 81.9 |
| | CaTS-SC | 81.5 | 81.4 | 81.5 | 81.7 | 81.9 | 81.8 | **82.1** |
| **Object_Counting** | CaTS-ES | 70.0 | 71.6 | 69.6 | 68.0 | **73.6** | 72.0 | 67.2 |
| | CaTS-ASC | **73.6** | 73.6 | 74.4 | 73.6 | **76.0** | 73.2 | 74.8 |
| | CaTS-SC | 72.8 | **74.0** | 73.2 | 72.4 | 74.4 | 73.6 | 72.8 |
| **ARC_challenge** | CaTS-ES | **86.8** | 86.4 | **86.8** | 86.5 | **86.8** | 86.4 | 86.2 |
| | CaTS-ASC | **86.7** | 86.6 | 86.6 | 86.6 | **86.7** | 86.6 | 86.6 |
| | CaTS-SC | 86.3 | 86.1 | 86.1 | **86.7** | 86.3 | 86.6 | 86.4 |

# F    RESULTS FOR DIFFERENT SAMPLE BUDGETS

Here, we show the performance under different sample budgets of other datasets and models.

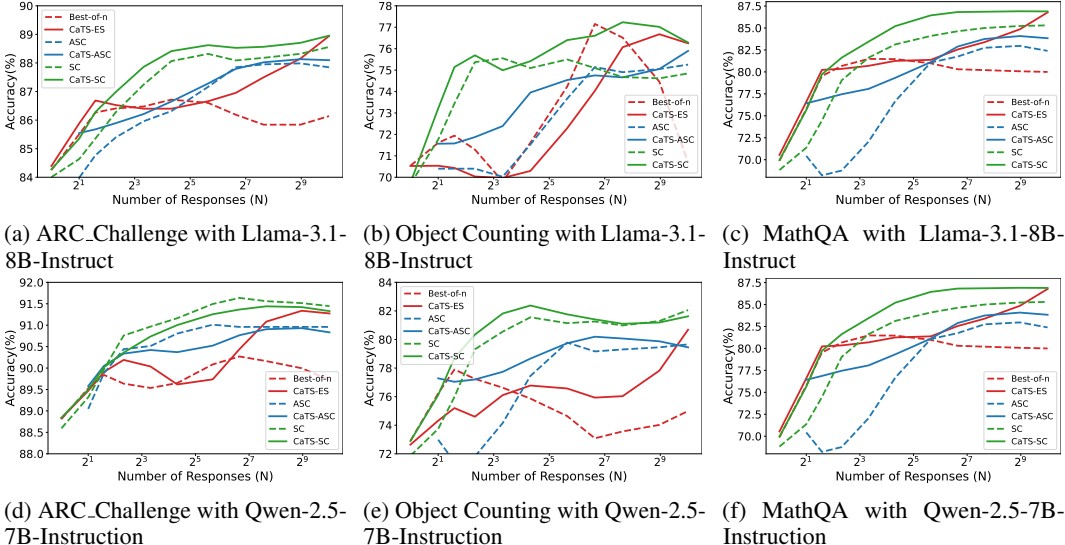

(a) ARC_Challenge with Llama-3.1-8B-Instruct

(b) Object Counting with Llama-3.1-8B-Instruct

(c) MathQA with Llama-3.1-8B-Instruct

(d) ARC_Challenge with Qwen-2.5-7B-Instruction

(e) Object Counting with Qwen-2.5-7B-Instruction

(f) MathQA with Qwen-2.5-7B-Instruction

Figure 4: Performance comparison of different inference strategies across datasets and models using CaTS.

# G    HYPERPARAMETERS

This section details the hyperparameters used in our experiments. We categorize them into training data generation, training process, and response generation

## G.1    TRAINING DATA GENERATION

When creating the datasets, we set the number of responses for each query $N = 32$. For the parameter in dynamic temperature, we follow the default hyperparameter settings from the original paper: $T_0 = 0.8$, $M = 0.8$, $\gamma = 1.0$, and $\tau_0 = 0.001$.

## G.2    TRAINING PROCESS

In the training objective, we set the threshold $\eta = 0.75$ to filter the response used in generation ability training and the weight $w = 0.1$ to balance two losses.

In the training process, we use the AdamW optimizer with a learning rate of $5 \times 10^{-5}$. The total number of training samples is set to 100,000, while 1,000 samples are used for evaluation. We employ a batch size of 1 with gradient accumulation steps of 64 to simulate a larger effective batch size. The model is trained for 1 epoch.

For parameter-efficient fine-tuning, we apply LoRA with rank $r = 32$, scaling factor $\alpha = 16$, and dropout rate of $0.05$. In the whole training examples, the ratio of causal language modeling data is 0.7. We train the model on multiple datasets with varying proportions of training and evaluation data. Specifically, GSM8K and SVAMP each contribute 15% of the training and evaluation samples. SciQ, CommonsenseQA, Winogrande, OpenBookQA, ReClor, ARC-Easy, and LogiQA each contribute 5% of the training and evaluation samples.

During the sample training data selection process, we ensure that the data is evenly distributed across different confidence intervals. This balancing strategy prevents overrepresentation of any specific confidence range, allowing the model to learn from a diverse set of samples. By maintaining an

| Methods | Llama-3.1-8B-Instruct | | | Qwen2.5-7B-Instruct | | | DeepSeek-R1-Distill-1.5B | | |
|---|---|---|---|---|---|---|---|---|---|
| | GPQA | Hellaswag | MMLU_pro | GPQA | Hellaswag | MMLU_pro | GPQA | Hellaswag | MMLU_pro |
| SC | 33.28 | 66.62 | 49.19 | 34.72 | 71.16 | 52.83 | 38.32 | 36.65 | 34.56 |
| Self-Certainty | 34.12 | 68.73 | 50.44 | 35.91 | 72.58 | 54.21 | 38.71 | 37.52 | 35.12 |
| CaTS-SC | **35.53** (+2.3) | **72.84** (+6.2) | **53.43** (+4.2) | **40.91** (+6.2) | **77.04** (+5.9) | **57.66** (+4.8) | **39.53** (+1.2) | **40.48** (+3.8) | **37.47** (+2.9) |
| Best-of-N | 32.26 | 67.28 | 47.31 | 37.24 | 71.02 | 51.87 | 37.16 | 36.97 | 32.63 |
| CaTS-ES | **36.55** (+4.3) | **73.72** (+6.4) | **53.83** (+6.5) | **40.40** (+3.2) | **76.72** (+5.7) | **58.18** (+6.3) | **39.53** (+2.4) | **40.60** (+3.6) | **37.68** (+5.1) |
| ASC | 33.79 | 66.52 | 49.37 | 34.72 | 71.22 | 52.53 | 39.49 | 36.47 | 34.42 |
| CaTS-ASC | **36.55** (+2.8) | **72.70** (+6.2) | **53.63** (+4.3) | **38.89** (+4.2) | **76.56** (+5.3) | **57.20** (+4.7) | **40.70** (+1.2) | **40.04** (+3.6) | **37.31** (+2.9) |
| ESC | 35.03 | 72.52 | 52.84 | 36.36 | 76.98 | 56.98 | 39.53 | 40.22 | 36.83 |
| RASC | 34.86 | 72.36 | 52.90 | 37.76 | 76.76 | 57.00 | 39.32 | 40.12 | 36.65 |

Table 10: The results for additional datasets.

equal number of training examples in each confidence bin, we improve the robustness of confidence calibration and reduce potential biases in the learning process.

### G.3 RESPONSE GENERATION

When generating the response, we set the temperature equals to 1.0.

## H EXPLANATION OF CALIBRATION ESTIMATION

ECE measures the discrepancy between a model's predicted confidence and its actual accuracy, defined as:

$$\text{ECE} = \sum_{m=1}^{M} \frac{|B_m|}{N} \left| \text{acc}(B_m) - \text{conf}(B_m) \right|,$$

where $M$ is the number of bins, $B_m$ represents the set of samples in the $m$-th bin, and $N$ is the total number of samples. A lower ECE value indicates better calibration, meaning the model's confidence aligns more closely with its actual correctness. In our experiments we set $m = 10$.

## I RESULTS ON ADDITIONAL DATASETS AND MODELS

We provide the results in three more dataset in Table 10 and Qwen3-4B(non-thinking) in Table 11.

| Method | Obj_C | MathQA | ARC_C | GPQA | Hellaswag | MMLU_Pro |
|---|---|---|---|---|---|---|
| SC | 87.60 | 86.04 | 91.29 | 36.87 | 70.88 | 57.90 |
| **CaTS-SC** | **89.15** (+1.55) | **87.85** (+1.81) | **92.54** (+1.25) | **38.52** (+1.65) | **72.40** (+1.52) | **59.65** (+1.75) |
| Best-of-N | 86.80 | 84.52 | 90.86 | 36.27 | 69.80 | 55.56 |
| **CaTS-ES** | **88.42** (+1.62) | **86.30** (+1.78) | **92.10** (+1.24) | **38.05** (+1.78) | **71.55** (+1.75) | **57.40** (+1.84) |
| ASC | 88.00 | 86.07 | 91.37 | 37.03 | 70.74 | 57.87 |
| **CaTS-ASC** | **89.80** (+1.80) | **87.95** (+1.88) | **92.72** (+1.35) | **38.90** (+1.87) | **72.60** (+1.86) | **59.95** (+2.08) |
| ESC | 86.80 | 86.08 | 91.37 | 37.53 | 70.84 | 57.99 |
| **CaTS-ESC** | **88.55** (+1.75) | **87.80** (+1.72) | **92.65** (+1.28) | **39.20** (+1.67) | **72.55** (+1.71) | **60.10** (+2.11) |

Table 11: Performance comparison of CaTS variants and baseline methods across multiple benchmarks. Numbers in parentheses indicate improvements over the corresponding baseline.

## J REWARD MODEL TIME COMPARE

When the reward model and generation model are similar in size, inference with both models doubles computational cost and latency, as each input requires two forward passes—one for generation and one for evaluation. In contrast, Self-Calibration integrates confidence estimation into the generation model, enabling single-pass inference and significantly improving efficiency. Empirically, we find that reward model evaluation takes 1.71 seconds per sample (without batching), while Self-Calibration requires only 1.08 seconds, showing the efficiency advantage of our method.

## K SUPERVISED CORRECTNESS VS. SELF-INDUCED CONFIDENCE

To further examine the effectiveness of our self-induced confidence signal, we conduct an ablation study in which the confidence supervision is replaced with binary correctness labels (0/1). The results are shown in Table 12. Across all benchmarks, supervised correctness provides only slight improvements over the baseline but consistently underperforms compared to the self-induced confidence signal. This suggests that the soft, self-consistency–derived supervision offers more informative guidance than discrete correctness labels. The limitations of supervised correctness are particularly evident in the early-stopping (ES) setting. When trained with binary labels, the model's confidence predictions collapse toward the extremes (0 or 1), which substantially degrades CaTS-ES performance. In contrast, the unsupervised confidence signal remains stable and continues to support reliable early stopping.

Overall, these findings demonstrate that directly supervising confidence with correctness labels does not yield performance gains. The self-induced confidence signal is both more informative and more robust, especially in scenarios involving early stopping and confidence-weighted voting.

| Method | Obj_C | MathQA | ARC_C |
|---|---|---|---|
| CaTS-SC | 76.8 | 83.6 | 87.7 |
| CaTS-SC (supervised) | 76.2 | 81.2 | 87.3 |
| CaTS-ES | 76.8 | 83.6 | 87.7 |
| CaTS-ES (supervised) | 69.8 | 81.2 | 86.2 |
| CaTS-ASC | 75.2 | 81.9 | 86.6 |
| CaTS-ASC (supervised) | 74.8 | 80.2 | 86.6 |

Table 12: Ablation results comparing self-induced confidence with supervised correctness across different CaTS variants.

## L    TOKEN NUMBER ANALYSIS

We report the average token usage in MathQA with our Llama-3.1-Instruct-self-calibration model, showing that the overall token-level count remains proportional to the sample count. Therefore, the reported efficiency gains remain valid and reflective of actual resource usage in practice.

| Response # | Self-Consistency # Token | CaTS-SC # Token |
|---|---|---|
| 1 | 345.28 | 355.28 |
| 2 | 690.99 | 710.87 |
| 3 | 1022.44 | 1051.99 |
| 5 | 1633.28 | 1681.15 |
| 10 | 2804.93 | 2890.71 |
| 20 | 3375.38 | 3482.24 |

Table 13: Token usage comparison between Self-Consistency and our method.

## M    ADDITIONAL RELATED WORK

Recent test-time scaling approaches also leverage intrinsic confidence signals, but differ fundamentally from our self-calibration framework. Deep Think with Confidence (Fu et al., 2025) extracts token- or step-level confidence to filter reasoning traces, whereas our method derives a soft, self-consistency–based confidence signal that reflects reasoning-level correctness. Thought Calibration (Wu et al., 2025) relies on trained probes over hidden states to detect reasoning errors, while our approach does not need access to hidden states and labelled data, and generates confidence directly from the model's own reasoning path. Guided by Gut (Ghasemabadi et al., 2025) enhances intrinsic confidence through RL and tree-search mechanisms and relies on an IsCorrect() function that required ground truth answers, while our lightweight, RL-free calibration yields a stable learnable confidence predictor for both voting and early stopping. Although surrogate confidence calibration has been explored for traditional classifiers (Le Coz et al., 2024), their binary calibration does not extend to multi-step reasoning; in comparison, our method performs soft self-calibration tailored to

LLM reasoning, providing a richer and more stable signal without collapsing to hard correctness labels than just labels.

