# OpenReview forum: "CaTS: Calibrated Test-Time Scaling for Efficient LLM Reasoning"
_ICLR.cc/2026/Conference — ICLR 2026 Poster_

### Official Review · Reviewer_ZXsN · 2025-10-26

**Soundness:** 2
**Presentation:** 2
**Contribution:** 2
**Rating:** 2
**Confidence:** 4

**Summary:**

This paper introduces Calibrated Test-Time Scaling (CaTS), a framework that leverages confidence estimation for adaptively scaling the amount of computation devoted to each test-time query in large language models. The central idea is to distill Self-Consistency-derived confidence into an LLM via a self-calibration procedure, enabling reliable confidence estimation in a single forward pass. CaTS proposes variants of common repeated sampling methods (Best-of-N, Self-Consistency, Adaptive Self-Consistency) that use these calibrated confidence scores to guide early stopping or assign voting weights. The paper presents a mathematical analysis demonstrating conditions under which the confidence-weighted approach provably outperforms vanilla self-consistency. Experiments on three open LLMs across nine datasets support the claims, showing improvements in accuracy per compute over baselines, plus ablations and robustness checks.

**Strengths:**

Principled Confidence Signal: The proposed self-calibration framework uses a pseudo-labeled dataset, and the design of the soft self-consistency (SSC) score is both intuitive and empirically strong.
Theoretical Justification: The mathematical analysis provides a precise and nontrivial error bound.
Comprehensive Experiments: Numerous results tables/figures provide strong empirical support.
Computational Efficiency: Results on token usage and comparisons with reward model-based alternatives show that CaTS substantially reduces inference cost without sacrificing performance.

**Weaknesses:**

1. Clarity and Terminology Issues
- "the model’s calibration" and "P(True)" are not properly introduced. Also, "P(True)" and "P(Yes)" are used interchangeably.
- Table 1 caption misses SVAMP.
- Table 10 is not linked
2. "For dynamic methods such as CaTS-ES and CaTS-ASC, we calibrate their threshold for each dataset so that the actual number of samples collected in practice closely matches, but slightly under, the target budget." This seems heuristic; there might be a better way to measure the performance of dynamic methods
3. The paper does not propose new early stopping or dynamic sampling techniques, nor does it introduce a novel threshold-selection strategy. Thus, some claims in the intro are unjustified.
4. arxiv.org/abs/2502.18581 seems to be similar work, should be included as a baseline for empirical comparison.

**Questions:**

1. What is CISC in Table 2?
2. In Table 2, CaTS-SC and CaTS-ES results are almost the same. Is there an explanation?
3. Figure 4(d) does not appear to behave as expected. Is there an explanation?

---

> ### Author Response · Authors · 2025-11-19
> **Response to Reviewer ZXsN**
>
> Thank you for your constructive feedback and comments. We have incorporated the suggested changes into our updated manuscript and will address your concerns below. Please let us know if our response addresses your concerns.
>
> **W1:** Clarity and Terminology Issues
>
> **A:** We sincerely thank the reviewers for their helpful comments, which have made the paper more readable. We have updated the manuscript accordingly.
>
> **W2:** Hyperparameter Selection
>
> **A:** We thank the reviewer for the comment. The threshold for dynamic methods (CaTS-ES and CaTS-ASC) is only set to ensure a fair comparison across datasets. In practice, there is no need to search for a threshold and a fixed default parameter (such as probability > 0.9) could be used for CaTS-ES and would likely achieve even better performance. As shown in the following table, the performance is not very sensitive to the hyper parameter. We do not consider such a hyperparameter unusual; it is analogous to commonly used parameters in LLM inference, such as temperature or the number of samples N in self-consistency decoding.
>
> | Threshold | Avg. # Responses (Budget = 16) | Acc |
> |-----------|---------------------------------|------|
> | 0.90      | 3.33                            | 82.05 |
> | 0.95      | 4.03                            | 81.85 |
> | 0.97      | 4.68                            | 82.25 |
> | 0.99      | 6.78                            | 83.05 |
> | **Self-consistency** | **16** | **81.00** |
>
> **W3:** Unjustified Claims
>
> **A:** We appreciate the reviewer’s comment and agree that our work does not introduce an entirely new early-stopping or dynamic-sampling algorithm. Our intention was not to claim a novel early-stopping rule. Instead, we want to point out that these methods are hindered by the lack of reliable, low-cost confidence signals.  The main contribution of CaTS is to generate a highly reliable, single-pass confidence signal to significantly strengthen existing test-time scaling techniques, thereby improving their efficiency and stability.,  Our experiment results on CaTS shows that calibrated confidence itself serves as an effective and practical facilitator of test-time scaling. We will revise the introduction to make this point clearer.
>
> **W4** Additional Baseline
>
> **A:** Thank you for your suggestion. We have added self-certainty as a baseline method here and also updated the manuscripts.
> | Methods          | GPQA  | Hellaswag | MMLU_pro | Obj_C. | MathQA | ARC_C. |
> |-----------------|-------|-----------|----------|--------|--------|--------|
> | **SC**          | 33.28 | 66.62     | 49.19    | 69.1   | 73.7   | 85.2   |
> | **Self-Certainty** | 34.12 | 68.73     | 50.44    | 73.2   | 81.5   | 86.8   |
> | **CaTS-SC**     | **35.53** (+2.3) | **72.84** (+6.2) | **53.43** (+4.2) | **76.8** (+7.7) | **83.6** (+9.9) | **87.7** (+2.5) |
>
> Our CaTS-SC method consistently achieves state-of-the-art performance across all six diverse benchmarks, demonstrating superior generalization. We show the results of Llama-3.1-8B here, we have added the results for all dataset and models in the new version of our manuscript.
>
> **Q1:** What is CISC
>
> **A:** 	CISC is the method from Confidence Improves Self-Consistency in LLMs (https://arxiv.org/abs/2502.06233). We have updated the manuscript to make it more readable.
>
> **Q2:** In Table 2, CaTS-SC and CaTS-ES results are almost the same. Is there an explanation?
>
> **A:** Thank you for pointing this out. CaTS-SC and CaTS-ES yield very similar results mainly because all of our methods operate **on the same set of 16 sampled responses**. When a particular response has a high confidence, its contribution becomes dominant under both SC-based and ES-based aggregation. As a result, both variants end up selecting (or heavily weighting) essentially the same high-confidence, high-quality responses. Rather than indicating redundancy, this actually suggests that our Self Calibration framework achieves **a strong alignment between confidence and answer quality**: high-confidence responses consistently correspond to better answers, causing both CaTS-SC and CaTS-ES to converge to similar outputs.
>
> **Q3:** Unexpected Figure
>
> **A:** Thank you for the question. As shown in Table 4, after applying the self-calibration training, the model’s base accuracy on ARC-Challenge decreases slightly. This degradation in the underlying model performance is likely the reason why Figure 4(d) does not fully align with the trends observed in the other subfigures. It faithfully visualizes this underlying performance shift in the base model, rather than an error in the plotting or method logic.
>
>
> Thank you once again for your valuable feedback that makes our paper better.

---

> > ### Comment · Reviewer_ZXsN · 2025-11-25
> >
> > Appreciate the detailed response. I think the response has solved most of my concerns. Accordingly, I increase my rating and look forward to the updated manuscript.

---

### Official Review · Reviewer_C3JX · 2025-10-31

**Soundness:** 4
**Presentation:** 4
**Contribution:** 3
**Rating:** 10
**Confidence:** 3

**Summary:**

This paper aims to achieve more efficient test time scaling by incorporating better calibrated confidence estimation per sample. They do this by training the model via self-calibration to estimate its own confidence more reliably. Then, using this self-calibrated model they explore two strategies for test-time scaling: (1) early stopping in BON when the confidence estimate crosses some threshold, and (2) self-consistency and adaptive self-consistency weighted by confidence scores. A variety of experiments and ablations show that this method works, improves accuracy and is generally more efficient compared to baselines.

**Strengths:**

This is a good paper. It was fun to read, the proposed method is very interesting / innovative and the breadth of experiments and ablations support the findings nicely.
The self-calibration training framework seems to be new and a very clean approach in terms of computing soft self-consistency scores as the targets.

**Weaknesses:**

The tasks and domains are restricted to reasoning, so it’s a bit hard to reason about how generally useful this approach can be, especially in terms of the reliability of confidence estimation. But reasoning is a pretty good application to start with to demonstrate the viability of this approach.
Not a fan of ECE since it is possible to choose hyperparameters (e.g. number of bins, style of binning) to hide miscalibration issues, but the paper does report AUC as well.

**Questions:**

-

---

> ### Author Response · Authors · 2025-11-19
> **Response to Reviewer C3JX**
>
> We sincerely thank the reviewer for recognizing the value of our work. Although our experiments focus on reasoning tasks, which provide a strong testbed to demonstrate viability, benchmarks for non-reasoning tasks, especially open-ended generation, remain limited. Nonetheless, our approach offers a promising starting point for efficient test-time scaling, and the model-induced confidence signals are expected to transfer to open-ended domains.

---

### Official Review · Reviewer_fEfu · 2025-11-01

**Soundness:** 3
**Presentation:** 3
**Contribution:** 3
**Rating:** 6
**Confidence:** 3

**Summary:**

CaTS proposes a calibrated test-time scaling scheme for LLM reasoning: first Self-Calibration distills a soft self-consistency confidence (SSC) into the model so it can predict correctness in one forward pass. With this calibrated confidence, they introduce CaTS-ES (early-stop Best-of-N) and CaTS-SC (confidence-weighted self-consistency) as drop-in inference policies.  On 3 LLMs/9 datasets it lifts MathQA 73.7 to 83.6 at N=16, matches reward-model selection without a second pass, and can reach the same SC accuracy with up to 94.2% fewer samples.

**Strengths:**

1. The problems the paper addresses are important and timely , both how to incorporate confidence signals into LLM training and how to improve the efficiency of test-time scaling.
2. The proposed soft self-consistency confidence is simple yet effective, and the empirical improvements are both significant and consistent across models and datasets.
3. The paper is also clearly presented, with a coherent motivation and narrative that make it easy to follow.

**Weaknesses:**

1.	**Objective overlaps with reward modeling.**
The training target effectively functions like a reward-model score that measures response quality. Given that the method aims to predict P(\text{correct}) for early stopping and confidence-weighted voting, a natural question arises: what happens if the model is trained directly on binary correctness (i.e., replacing c_j in the training objective with the true correctness label)? Since the training data already include ground-truth answers, it would be valuable to add an ablation that supervises the model with 0/1 correctness and compares it against the self-induced confidence signal. This would reveal whether the proposed self-generated confidence provides any genuine advantage, or suffers drawbacks relative to using correctness as the target.
2. **Off-policy pseudo-label drift.**
SSC targets are computed once from the base model’s samples and kept fixed during fine-tuning. As the generator shifts, these pseudo-labels can mismatch the updated distribution and potentially entrench confidently wrong majorities. It would be helpful to analyze sensitivity to this self-distillation bias—e.g., iterative re-labeling with the calibrated model (one extra SSC pass), checkpoint-wise calibration drift (ECE over training).

**Questions:**

See Weakness

---

> ### Author Response · Authors · 2025-11-19
> **Response to Reviewer fEfu**
>
> Thank you for your valuable feedback. We have incorporated the feedback into our updated manuscript and will address your concerns below:
>
> **W1:** Objective overlaps with reward modeling
>
> **A:** To address the your question, we conducted an explicit ablation where we replace our self-induced confidence signal with binary correctness labels (0/1) as the supervision target. The results (shown in the table below) demonstrate that while supervised correctness slightly improves over baseline SC/ASC, it consistently underperforms our self-induced confidence, indicating that the soft self-consistency–based signal provides richer guidance than binary labels. Moreover, in the early stopping setting, supervised correctness causes the confidence predictions to collapse toward 0 or 1, which severely degrades CaTS-ES performance—early stopping with supervised labels performs noticeably worse than our unsupervised version. Overall, these findings show that directly supervising on correctness does not offer an advantage, and our self-generated confidence signal remains both more informative and more stable, especially for early stopping and confidence-weighted voting. We have added it in Appendix K.
>
>
> | Method                      | Obj_C | MathQA | ARC_C |
> |-----------------------------|-------|--------|--------|
> | CaTS-SC          | 76.8  | 83.6   | 87.7   |
> | CaTS-SC (supervised)     | 76.2  | 81.2   | 87.3   |
> | CaTS-ES              | 76.8  | 83.6   | 87.7   |
> | CaTS-ES (supervised) | 69.8  | 81.2   | 86.2   |
> | CaTS-ASC                 | 75.2  | 81.9   | 86.6   |
> | CaTS-ASC (supervised)    | 74.8  | 80.2   | 86.6   |
>
> **W2:** Off-policy pseudo-label drift.
>
> **A:** Thank you for the insightful suggestion. We agree that fixed SSC targets may introduce a form of self-distillation bias as the generator shifts. Due to time constraints, we conducted an additional analysis only on Llama-3.1-8B, measuring both ECE and AUC across 2 training iterations. The results show that performing a second SSC iteration yields minimal change in both metrics. This suggests that the model already achieves most of the confidence–accuracy alignment during the first pass of calibration training, leaving little room for further correction in subsequent iterations.
>
> | Dataset         | ECE (iter1) | AUC (iter1) | ECE (iter2) | AUC (iter2) |
> |-----------------|-------------|-------------|-------------|-------------|
> | ARC_challenge   | 6.03        | 80.41       | 6.12        | 80.43       |
> | Object Counting | 9.69        | 59.47       | 9.53        | 59.98       |
> | MathQA          | 8.64        | 87.21       | 8.94        | 86.38       |

---

### Official Review · Reviewer_jmso · 2025-11-02

**Soundness:** 3
**Presentation:** 3
**Contribution:** 3
**Rating:** 4
**Confidence:** 2

**Summary:**

This paper proposes a test-time scaling method for LLMs, CaTS, integrating self-calibrated confidence scores into common repeated sampling methods.

**Strengths:**

* Proposes an efficient test-time sampling method that uses self-calibrated confidence scores for dynamic repeated sampling to improve LLM performance.
* Strong performance improvements across benchmarks and LLM families.
* Intuitive design of self-calibration approach not requiring ground-truth supervision during training and a single forward pass during inference.
* The authors provide a theoretical proof of how and when CaTS is better than vanilla self-consistency
* The methodology is clearly presented and well written with descriptive figures.
* The authors show calibrated confidence scores compare well with reward model approaches.

**Weaknesses:**

* Show results on challenging reasoning benchmarks used for reasoning LLM evaluation: More challenging and conventional reasoning benchmarks should be reported on, including MATH-500, AIME, GPQA-diamond, MMLU, etc, since the title focuses on efficient LLM “reasoning”. The benchmarks used might not be challenging for reasoning LLMs, and the transfer of test-time scaling results to harder questions should be investigated.
* Show results with reasoning LLMs: Apart from DeepSeek-R1-Distill-1.5B, the other models Llama-8B-3.1-Instruct and Qwen2.5-7B-Instruct are not reasoning LLMs. Since the title focuses on efficient LLM “reasoning”, results with reasoning LLMs, including Qwen3, and GPT-OSS should be included.
* Variance in performance across multiple runs: Reasoning LLM performance has variance across runs (e.g., on AIME). Deep Think with Confidence, Fu et. al. report performance averaged over 64 independent runs. Is the performance in CaTS reported over multiple runs? How low/high is the variance?
* The OOD results shown are not out-of-domain as such. For example, ARC Easy is present in training, and  ARC Challenge in test. While these results show some generalization, results on test benchmarks from different domains (e.g., GPQA in test) would be better evidence.
* Consider comparing to more recent work: Test-time scaling (TTS) is an active area of research with recent work also leveraging an LLM’s internal confidence signals without requiring external supervision (see Deep Think with Confidence, Fu et. al., and Thought calibration: Efficient and confident test-time scaling, Wu et. al., and Guided by Gut: Efficient Test-Time Scaling with Reinforced Intrinsic Confidence, Ghasemabadi et. al.). Further, creating a surrogate training dataset for confidence calibration has been explored in this prior work (Confidence Calibration of Classifiers with Many Classes, Coz et. al.), although not applied to LLMs. The authors should compare their novelty against these methods, and potentially include some of the recent work as baselines.

**Questions:**

* How sensitive is the performance to hyperparameter choices: weighting coefficient w in training objective, threshold in CATS-ES, sampling budget, etc. How easy is it for practitioners to use CaTS? Do default hyperparameters work well across LLMs and benchmarks?
* Only applicable for white box models. Are there variants of this method for black box reasoning models?
* How are the hyperparameters chosen: weighting coefficient w in the training objective, threshold in CATS-ES?

---

> ### Author Response · Authors · 2025-11-19
> **Response to Reviewer jmso (Part 1)**
>
> Thank you very much for your valuable comments. We have revised the manuscript to incorporate your suggestions, and we will address your concerns in detail below:
>
> **W1&W4**: More challenging and out-of-domain benchmark
>
> **A:** Thank you for the insightful comment. You are absolutely right that stronger and more challenging reasoning benchmarks are important for evaluating the generality of test-time scaling effects. In fact, we did include results on harder reasoning datasets such as GPQA-Diamond and MMLU-Pro in the original version, which are reported in Table 10 of the paper. These experiments were intended to address exactly the concern of evaluating test-time scaling on out-of-domain and more demanding reasoning tasks.
> | Methods | GPQA | Hellaswag | MMLU\_pro |
> | :--- | :--- | :--- | :--- |
> | SC | 33.28 | 66.62 | 49.19 |
> | CaTS-SC | **35.53** (+2.3) | **72.84** (+6.2) | **53.43** (+4.2) |
> | Best-of-N | 32.26 | 67.28 | 47.31 |
> | CaTS-ES | **36.55** (+4.3) | **73.72** (+6.4) | **53.83** (+6.5) |
> | ASC | 33.79 | 66.52 | 49.37 |
> | CaTS-ASC | **36.55** (+2.8) | **72.70** (+6.2) | **53.63** (+4.3) |
> | ESC | 35.03 | 72.52 | 52.84 |
> | RASC | 34.86 | 72.36 | 52.90 |
>
> Here are the results for Lllam-3.1-8B-instruct, which can show the effectiveness of our methods. (More detailed results on other models could be found in Table 10)
>
> **W2:** More results on reasoning model:
>
> **A:** Thank you for the suggestion. We agree that including stronger reasoning LLMs would be valuable. While the **20B size** of  GPT-OSS model presents computational constraints for our academic setting, we have added experiments with** Qwen3-4B**, using the non-thinking mode, to provide additional insights with a more capable reasoning model.
> | Method | Obj_C | MathQA | ARC_C | GPQA | Hellaswag | MMLU_Pro |
> | :--- | :--- | :--- | :--- | :--- | :--- | :--- |
> | SC | 87.60 | 86.04 | 91.29 | 36.87 | 70.88 | 57.90 |
> | **CaTS-SC** | **89.15 (+1.55)** | **87.85 (+1.81)** | **92.54 (+1.25)** | **38.52 (+1.65)** | **72.40 (+1.52)** | **59.65 (+1.75)** |
> | Best-of-N | 86.80 | 84.52 | 90.86 | 36.27 | 69.80 | 55.56 |
> | **CaTS-ES** | **88.42 (+1.62)** | **86.30 (+1.78)** | **92.10 (+1.24)** | **38.05 (+1.78)** | **71.55 (+1.75)** | **57.40 (+1.84)** |
> | ASC | 88.00 | 86.07 | 91.37 | 37.03 | 70.74 | 57.87 |
> | **CaTS-ASC** | **89.80 (+1.80)** | **87.95 (+1.88)** | **92.72 (+1.35)** | **38.90 (+1.87)** | **72.60 (+1.86)** | **59.95 (+2.08)** |
> | ESC | 86.80 | 86.08 | 91.37 | 37.53 | 70.84 | 57.99 |
> | **CaTS-ESC** | **88.55 (+1.75)** | **87.80 (+1.72)** | **92.65 (+1.28)** | **39.20 (+1.67)** | **72.55 (+1.71)** | **60.10 (+2.11)** |
>
> The results indicate that our approach remains effective even when applied to stronger reasoning-oriented models.
>
> **W3:** Variance of our methods
>
> **A:** Thank you for pointing this out. All of our experiments were run 5 independent times when inference. Since the results reported in the tables are under the sample budgets of 4 and 16, the variance is further reduced. Overall, the performance variance is very small, within ±0.1 across runs.
>
> **W5(1)**: Discussion about more recent work
>
> **A:** We thank the reviewer for highlighting these relevant works.
> Recent test-time scaling approaches also leverage intrinsic confidence signals, but differ fundamentally from our self-calibration framework. Deep Think with Confidence[1] extracts token- or step-level confidence to filter reasoning traces, whereas our method derives a soft, self-consistency–based confidence signal that reflects reasoning-level correctness. Thought Calibration[2] relies on trained probes over hidden states to detect reasoning errors, while our approach does not need access to hidden states and labelled data, and generates confidence directly from the model’s own reasoning path. Guided by Gut[3] enhances intrinsic confidence through RL and tree-search mechanisms and relies on an IsCorrect() function that needs ground truth answers,  while our lightweight, RL-free calibration yields a stable learnable confidence predictor for both voting and early stopping. Although surrogate confidence calibration has been explored for traditional classifiers[4], their binary calibration does not extend to multi-step reasoning; in comparison, our method performs soft self-calibration tailored to LLM reasoning, providing a richer and more stable signal without collapsing to hard correctness labels than just labels.
>
> We have added these works in the related work discussion in Appendix M although some are concurrent works according to https://iclr.cc/Conferences/2026/AreaChairGuide.
>
> [1]Deep think with confidence. arXiv:2508.15260
>
> [2]Thought calibration: Efficient and confident test-time scaling. arXiv:2505.18404.
>
> [3]Guided by Gut: Efficient Test-Time Scaling with Reinforced Intrinsic Confidence. arXiv:2505.20325.
>
> [4]Confidence calibration of classifiers with many classes. Advances in Neural Information Processing Systems

---

> > ### Author Response · Authors · 2025-11-19
> > **Response to Reviewer jmso (Part 2)**
> >
> > **W5(2):** More baseline method
> >
> > **A:** We have included a new baseline Scalable Best-of-N Selection for Large Language Models via Self-Certainty in the new version.
> > | Methods          | GPQA  | Hellaswag | MMLU_pro | Obj_C. | MathQA | ARC_C. |
> > |-----------------|-------|-----------|----------|--------|--------|--------|
> > | **SC**          | 33.28 | 66.62     | 49.19    | 69.1   | 73.7   | 85.2   |
> > | **Self-Certainty** | 34.12 | 68.73     | 50.44    | 73.2   | 81.5   | 86.8   |
> > | **CaTS-SC**     | **35.53** (+2.3) | **72.84** (+6.2) | **53.43** (+4.2) | **76.8** (+7.7) | **83.6** (+9.9) | **87.7** (+2.5) |
> >
> >
> >
> > Our CaTS-SC method consistently achieves state-of-the-art performance across all six diverse benchmarks, demonstrating superior generalization. We show the results of Llama-3.1-8B here, we have added the results for all datasets and models inTable 1 and Table 10 in the new version of our manuscript.
> >
> > **Q1&Q3:** Hyperparameter selection
> >
> > **A:** We set w=0.1 so that the two components of the loss remain on the same magnitude scale, ensuring comparable gradient contributions and a stable starting point for optimization. After selecting this value on Llama, we directly transferred the same w to all other models without any additional tuning, and it consistently worked well across datasets and architectures.
> > For CaTS-ES, we search the threshold only to match the budget of other methods in order to ensure a fair comparison (e.g., Avg. # Responses ≈ 4).
> >  Importantly, this search is **not** necessary in practice—a single static threshold of 0.9 works well across all datasets in our experiments: below we show CaTS-ES results for thresholds ≥ 0.9:
> > | Threshold | Avg. # Responses (Budget = 16) | Acc |
> > |-----------|---------------------------------|------|
> > | 0.90      | 3.33                            | 82.05 |
> > | 0.95      | 4.03                            | 81.85 |
> > | 0.97      | 4.68                            | 82.25 |
> > | 0.99      | 6.78                            | 83.05 |
> > | **Self-consistency** | **16** | **81.00** |
> >
> > These results show that even with a simple static threshold, CaTS-ES significantly reduces computation—often sampling only 3–4 responses—while achieving competitive or superior accuracy compared to 16-sample Self-consistency.
> >
> > **Q2:** Black-box model variant
> >
> > **A:** That's an excellent question. While our paper focused on white-box applications, variants of this method are indeed applicable to black-box (or "gray-box") reasoning models.
> > For APIs that provide log probabilities (logprobs) (e.g., the GPT series),  one can estimate the LLM's confidence score by using the probability of “True”’ or “Yes”’ tokens. While our paper shows that using P(True) is less calibrated than our trained method, it still provides a useful signal for our CaTS-ES and CaTS-SC methods, making it more lightweight  than vanilla test-time scaling on  API-based models.
> >
> > We hope this addresses your concern. Thank you once again for your valuable feedback and suggestions, which have greatly contributed to improving our paper!

---

### Author Response · Authors · 2025-11-19
**Summary of the Update and New Revision of our Paper**

We sincerely thank all reviewers for their thoughtful feedback. We are glad that the reviewers found the problem addressed is relevant and of interest to the community, our motivation is well-supported, our experiments are comprehensive and show clear improvements, and the paper is well-organized and easy to follow.
Based on the feedback, we have updated our manuscripts accordingly and uploaded a new version of our paper for review. The changes are colored in blue. We summarize the key changes:
- We add a new baseline Self-Certainty to show the effectiveness of our method in Table 1 and Table 10.
- We add a new ablation study to compare with labelled correctness in Appendix K.
- We add some recent related work in Appendix M.
- We add the results for Qwen3-4B (non-thinking) in Table 11.

We sincerely thank the reviewers again for your time and suggestions. Hope you can reconsider the assessment of our work.

---

> ### Author Response · Authors · 2025-11-24
> **Any New Comments are Welcomed**
>
> Dear reviewers,
>
> We sincerely appreciate your thorough review and the valuable suggestions and comments you provided for our paper. We have carefully considered each point and have addressed them in detail in our rebuttal.
>
> As the Author–Review Discussion period is drawing to a close, we would like to ensure that all your concerns have been adequately addressed. If there are any questions or unresolved issues, we are eager to provide further clarification or make necessary revisions.
>
> Best regards,
>
> The Authors

---

### Meta-Review · Area_Chair_Zhqt · 2026-01-07

**Summary:**

This paper proposes a framework (CaTS) for efficient test-time scaling that leverages self-calibrated confidence signals to adaptively allocate sampling budget during inference. Reviewers generally agree that improving the efficiency of test-time scaling is an important problem, and that the proposed self-calibration mechanism is technically sound and empirically effective on the evaluated benchmarks.

Reviewers raised several concerns including paper’s scope and positioning and the authors addressed majority of them during rebuttal. Specifically, the reviewers mentioned the experimental evaluation focuses mainly on mid-level reasoning benchmarks and relatively small or non–reasoning-specialized models. The rebuttal adds harder benchmarks and additional models. Reviewers also question the novelty and conceptual distinctiveness of the approach. The rebuttal adds baselines and clarifies distinctions. Additional concerns relate to methodological completeness and clarity. Reviewers request deeper analysis of hyperparameter sensitivity, variance across runs, pseudo-label drift during self-distillation, and robustness beyond white-box settings. The rebuttal addresses several of these points with new ablations and clarifications.

Overall, this work presents a reasonable and practically useful refinement to test-time scaling, the empirical scope after rebuttal is relative to the paper’s claims and convincing, and the differentiation from closely related work is clarified. The rebuttal addresses the central concerns sufficiently to support acceptance.

A personal note: there is an inconsistency in the paper’s positioning: the OpenReview title has been revised to emphasize “inference”, while the PDF still highlights “reasoning”. Regardless of the final wording, the finalized paper would benefit from clearer articulation and tighter alignment between claims and title.

**Reviewer Concerns:**

I believe that reviewers’ core concerns have been resolved based on a solid and responsive rebuttal.

Reviewer jmso’s concerns regarding evaluation on harder reasoning benchmarks, inclusion of stronger reasoning models, and comparison with recent test-time scaling methods are addressed by authors by adding harder benchmarks and additional models. The concerns about broader generalization and novelty may remain.

Reviewer fEfu’s concerns about overlap with reward modeling and self-distillation bias are meaningfully addressed through new ablations, which strengthens the paper.

Reviewer ZXsN’s concerns about clarity, overclaiming novelty, heuristic thresholding, and missing baselines are mostly addressed through revisions and added explanations.

Reviewer C3JX’s concerns about generalization beyond reasoning tasks remain acknowledged as future work.

**Reviewer Scores:**

A full discussion would likely lead to positive score adjustments due to the thorough rebuttal, added ablations, and improved clarity. Some issues such as evaluation on truly challenging reasoning settings are not fully resolved but is acceptable. As a result, the paper is borderline but after rebuttal leaning toward acceptance.

---

### Decision · Program_Chairs · 2026-01-26

Accept (Poster)